# Reverse Onco-Cardiology: What Is the Evidence for Breast Cancer? A Systematic Review of the Literature

**DOI:** 10.3390/ijms242216500

**Published:** 2023-11-19

**Authors:** Ioannis Boutas, Adamantia Kontogeorgi, Sophia N. Kalantaridou, Constantine Dimitrakakis, Panagiotis Patsios, Maria Kalantzi, Theodoros Xanthos

**Affiliations:** 1Breast Unit, Rea Maternity Hospital, 383 Andrea Siggrou Ave., Paleo Faliro, 175 64 Athens, Greece; 2Medical School, University of Crete, 13 Andrea Kalokairinoy Ave., 715 00 Giofirakia, Greece; 3Third Department of Obstetrics and Gynecology, Attikon University Hospital, Medical School, National and Kapodistrian University of Athens, 1 Rimini Str., 124 62 Chaidari, Greece; sophiakalantaridou@gmail.com; 4First Department of Obstetrics and Gynecology, Alexandra University Hospital, Medical School, National and Kapodistrian University of Athens, 4-2 Lourou Str., 115 28 Athens, Greece; dimitrac@ymail.com; 5Cardiology Department, Elpis General Hospital, 7 Dimitsana Str., 115 22 Athens, Greece; gsakka73@gmail.com; 6Post Graduate Study Program “Cardiopulmonary Resuscitation”, Medical School, National and Kapodistrian University of Athens, 75 Mikras Asias Str., 115 27 Athens, Greece; mariamkalan92@gmail.com; 7School of Health Sciences, University of West Attica, 28 Aghiou Spyridonos Str., 122 43 Aigaleo, Greece; txanthos@uniwa.gr

**Keywords:** breast cancer, reverse onco-cardiology, cardiovascular diseases, patient care

## Abstract

Breast cancer and cardiovascular diseases (CVD) represent significant global health challenges, with CVD being the leading cause of mortality and breast cancer, showing a complex pattern of incidence and mortality. We explore the intricate interplay between these two seemingly distinct medical conditions, shedding light on their shared risk factors and potential pathophysiological connections. A specific connection between hypertension (HTN), atrial fibrillation (AF), myocardial infarction (MI), and breast cancer was evaluated. HTN is explored in detail, emphasizing the role of aging, menopause, insulin resistance, and obesity as common factors linking HTN and breast cancer. Moreover, an attempt is made to identify the potential impact of antihypertensive medications and highlight the increased risk of breast cancer among those women, with a focus on potential mechanisms. A summary of key findings underscores the need for a multisystem approach to understanding the relationship between CVD and breast cancer is also explored with a highlight for all the gaps in current research, such as the lack of clinical observational data on MI and breast cancer in humans and the need for studies specifically designed for breast cancer. This paper concludes that there should be a focus on potential clinical applications of further investigation in this field, including personalized prevention and screening strategies for women at risk. Overall, the authors attempt to provide a comprehensive overview of the intricate connections between breast cancer and cardiovascular diseases, emphasizing the importance of further research in this evolving field of cardio-oncology.

## 1. Introduction

Breast cancer and cardiovascular disease (CVD) are among the most significant worldwide health challenges and the leading causes of mortality, respectively, according to a plethora of epidemiological data. Globally, 20.05 million CVD-related deaths occurred in 2021, a significant increase from the 12.1 million CVD deaths recorded in 1990 [1]. On the other hand, the mortality rate for breast cancer has fallen by about 40% over the past 20 years, while incidence rates have increased. The rate increased by 0.5% annually throughout the most recent data years (2010–2019), primarily due to localized-stage and hormone receptor-positive disease [2].

Cancer and cardiovascular disease are typically seen as two separate medical conditions. The obvious meeting point of the two pathological entities is cardiac neoplasms [3]. Most cardiac neoplasms are benign morphologies found in 0.0017–0.03% of autopsy series. Malignant neoplasms are usually metastatic but equally rare [4]. On the other hand, oncological patients show cardiovascular complications that either appear independently of their malignancy or are a result of the cardiotoxicity of the interventions made in the treatment of the neoplasm. Cardiologists who treat patients with cardiovascular diseases due to anticancer therapy (cardio-oncologists) are the primary contributors to the understanding that cancer and cardiovascular disease may coexist [5]. As can be deduced from the above, it is common for the majority of studies to focus either on the cardiotoxicity of administered chemotherapy regimens or on the incidence of CVD in the group of cancer patients. Thus, a new field termed cardio-oncology was established as a result of the dramatic rise in new cancer therapies and the significant rise in cancer survivorship [6]. This cardiology subspecialty develops primary and secondary risk approaches, as well as interventions, to stratify and reduce cardiovascular risk, prevent cardiovascular toxicity and its progression, and manage the adverse effects of anticancer treatments [7].

However, even though the causal relationship between cancer and cardiovascular disease has been extensively studied, potential linkages between current cardiovascular diseases and eventual malignancy are less well understood. Nevertheless, mainly observational studies led to the epidemiological conclusion that sometimes a cardiovascular event precedes the appearance of an independent malignancy. The first studies focused on colon and lung cancer, followed by cases of melanoma, which were extensively studied [8]. Now the participation of the Mendelian model methods suggests that people with cardiovascular disease have a higher risk of developing cancer at 17 sites than the general population, while prostate cancer seems to have a greater correlation [9]. As a result, “reverse cardio-oncology” has gained traction, as has a request for the greater clinician knowledge of the elevated cancer risk in patients with cardiovascular illness [10,11]. Obesity, diabetes mellitus, drunkenness, and cigarette use have been proposed as mutual risk factors for these two disease entities, which may explain, at least in part, the concurrent manifestation [12,13,14].

Furthermore, various ancillary processes and pathways related to cardiovascular illness have been implicated in cancer etiology. As a result, additional research is required to confirm and characterize the overlapping pathophysiological connections between cardiovascular disease and cancer. For the time being, practitioners should be aware of the elevated risk and provide recommendations and guidelines for the early detection of malignancy, particularly among patients with cardiovascular illness.

The case of breast cancer has so far not been studied regarding its possible relationship with reverse onco-cardiology. In addition to the existence of multiple common risk factors, it is possible that the immune failure that arises after a cardiovascular event may be related to the molecular mechanisms that mainly cause inflammatory breast cancer. This work aims at the meticulous collection and synthesis of all breast-related data and is the first systematic review of the subject.

## 2. Methods

The systematic review presented in this article adheres to the established PRISMA (Preferred Reporting Items for Systematic Reviews and Meta-Analyses) guidelines to ensure rigorous and comprehensive analysis. In pursuit of understanding the link between cardiovascular health and breast cancer, our research methodology encompassed the following steps.

### 2.1. Literature Search

We conducted an extensive search of pertinent medical databases over the past decade, specifically targeting studies related to the intersection of breast cancer and cardiovascular health. Our search was conducted in the two biggest medical databases: PubMed and Cochrane Library.

### 2.2. Keywords

To optimize the search for relevant studies, a comprehensive list of keywords was used. These included: “breast cancer”, “cardiovascular disease,” “onco-cardiology”, “cardiovascular risk factors”, “cardiotoxicity”, “treatment-related cardiotoxicity”, “chemotherapy-induced cardiotoxicity”, “radiotherapy”, “breast cancer survivors”, “heart disease”, “heart failure”, and “cardiovascular interventions”.

### 2.3. Query

The search query utilized across the databases was formulated as follows: (“breast cancer” OR “cardiovascular disease” OR “onco-cardiology”) AND (“cardiovascular risk factors” OR “cardiotoxicity” OR “treatment-related cardiotoxicity” OR “chemotherapy-induced cardiotoxicity” OR “radiotherapy” OR “breast cancer survivors” OR “heart disease” OR “heart failure” OR “cardiovascular interventions”).

### 2.4. Inclusion and Exclusion Criteria

Following the initial search, we employed a snowball procedure to select relevant papers. Only studies directly relevant to the interplay between breast cancer and cardiovascular health were included in our analysis. We did not take into account articles that study common risk factors, studies on the correlation of lifestyle, obesity, and cigarette smoking. These encompassed both original research papers and review articles. Only articles written in English were considered by the authors. Moreover, only the most recent studies were included in the final sample; more specifically, we did not include studies before 2000.

### 2.5. Reference Expansion

To ensure the comprehensiveness of our review, we further expanded our dataset by including papers referenced within the selected review articles. This process culminated in a total of twenty six research papers, all focused on the intricate relationship between breast cancer and cardiovascular health.

## 3. Results and Discussion

The result set consisted of 58,835 studies. All duplicate records were excluded. The remaining 6412 studies were independently assessed by three authors via an abstract review. For the relevant articles, the authors read the contents of each paper and all inclusion and exclusion criteria were applied to narrow down the result set to only the relevant studies of interest. For disagreements between the three reviewers, a consensus was required in order to resolve them. Moreover, the snowball procedure was followed to possibly identify more relevant studies. A total of 26 studies were the final result set, which was used for data extraction, as can be seen in Figure 1.

From the search in the literature, we concluded that it is more comprehensible if we group the studies based on the cardiovascular event to which they refer and which we consider as a causative factor in the occurrence of breast cancer. Thus, three subcategories emerged: studies focusing on arterial hypertension, atrial fibrillation, and finally, myocardial infarction. Also, as mentioned in defining the inclusion criteria of the study, we did not choose to refer individually to those studies that investigate the common risk factors between breast cancer and cardiovascular events, mainly because we thought that we would introduce bias into the sample of studies to be examined because the common factors, on the one hand, suggest some broader common pathogenetic mechanisms, and on the other hand, do not show specificity for a specific pathological entity, with the result that the study would lose the point on which it focuses, which is the clearly established cardiovascular events and their causal relationships with cancer breast.

### 3.1. Hypertension and Breast Cancer

Hypertension (HTN) is one of the most critical public health issues, and both health professionals and patients frequently concentrate on how seriously it affects the cardiovascular and cerebrovascular systems [15]. It is important to remember that HTN is also linked to the development of tumors in several organs, with the kidneys being particularly affected [16]. HTN is closely linked to both the incidence of breast cancer and the severity of the existing illness in the population of postmenopausal women [17]. Numerous observational studies evaluating the pathophysiological and clinical connection of the two diseased entities have been carried out since the beginning of this field of study in 1989 [18].

When examining the clinical data, it can be observed that numerous pathophysiological pathways overlap between the two entities. First, they are causally related to aging and menopause. Women over 55 are more likely to develop HTN [19] and BC [20]. Then, insulin resistance, type 2 diabetes, metabolic syndrome, and, ultimately, the development of HTN, are all closely related to the hormonal status of menopausal patients, notably estrogen withdrawal. Except for women who possess the KRAS variation [21], there is no obvious relationship between breast cancer risk and estrogen reduction; however, it is linked to aging and the postmenopausal era [20]. Obesity (BMI > 25 kg/m^2^) is another clear causal link between the two. Obesity is a major risk factor for HTN, and at the same time, it increases the risk of hormone-dependent breast tumors because adipose tissue is a hormone producer, which keeps estrogen levels high during menopause and maintains the regime of chronic inflammation, the activation of inflammatory phagocytes, tissue damage, and tumorigenesis in the angiogenic breast [22]. The chronic inflammatory state brought on by HTN, according to an immunological perspective, affects cellular programming systematically, inhibiting apoptosis and disrupting the control of cell turnover [23].

The possibility of several antihypertensive medications having an oncogenic effect is another aspect that indirectly links HTN with breast cancer. Researchers discovered that β-blockers may operate on receptors linked to processes that cause carcinogenesis, angiogenesis, and tumor spread to exercise their anti-tumor effects. Propranolol is a β-blocker with the ability to interfere with angiogenesis and modify the expression and activation of angiogenic signaling pathways, such as angiopoietin/TIE2, VEGF, and hypoxia-inducible factor, may also relieve BC cells [24]. The link between β-blockers and cancer development could be because β-adrenergic signaling has been found to regulate multiple cellular processes that contribute to the initiation and progression of cancer, including inflammation, angiogenesis, apoptosis, cell motility, and trafficking, and the activation of tumor-associated viruses, DNA damage repair, cellular immune response, and epithelial–mesenchymal transition (Cole SW). Finally, the data on calcium channel blockers (CCBs) that we obtained are conflicting; however, in the newer studies they seem to be unrelated [25].

The intersection of pathophysiological mechanisms is also confirmed clinically by multiple cohort and observational studies associating HTN with breast cancer. From 1998 to the present, multiple observational studies have been conducted in humans, of which twenty reached a statistically significant result that hypertension is associated with the occurrence of breast cancer, nine exclusively studied hypertension as a risk factor [26,27,28,29,30,31,32,33,34], one study suggested that normal–high diastolic blood pressure (DBP) values [35] are associated with increased BC risk in postmenopausal women, and the rest concerned the oncogenic effect of β-blockers in the breast [24,36,37,38,39,40,41,42,43,44,45]. Regarding CCBs, a recent meta-analysis of the data concluded that the two pathological entities are not correlated [25].

In the context of analyzing the association of arterial hypertension with the occurrence of breast cancer, we carried out a meta-analytical approach to the studies. Thus, we included eight studies from the USA, Brazil, Uruguay, Turkey, Korea, and China with a total number of 37,302 participants, of whom 15,746 developed breast cancer after establishing chronic hypertension. These studies were designed for multivariate observations and evaluated other factors as well. Data mining was carried out exclusively for hypertension and only in studies that reported a history of chronically established hypertension before the onset of breast cancer. While meta-analytically processing the results, it was found that the results were statistically significant regarding the existence of a correlation between hypertension and breast cancer; however, on the one hand, due to the limited number of studies and, on the other hand, because the studies were not designed to exclusively study hypertension as a causative factor in cancer, we do not believe that this result can be considered a strong statistical conclusion (Figure 2).

### 3.2. Atrial Fibrillation and Breast Cancer

With a global frequency of around 4%, atrial fibrillation is the most prevalent arrhythmia and a significant contributor to cardiovascular disease-related mortality [46]. Atrial fibrillation (AF) incidence is strongly correlated with age so the increase in human life expectancy makes atrial fibrillation more common [46]. Epidemiological studies suggest that women with atrial fibrillation had a higher risk of developing breast cancer, and a very recent meta-analysis from 2023 claims that among women with AF, 18% have an increased risk of developing BC without being able to prove a causal relationship [47].

There are no studies that are solely focused on the pathophysiological pathways that link AF with BC. One potential mechanism is that thrombin has tumorigenic characteristics in the breast. Endothelial dysfunction and injury, in particular, cause alterations in the extracellular matrix and interruption of nitric oxide production in AF. Abnormal changes in blood coagulation and hemostasis eventually emerge, as evidenced by higher prothrombotic blood indices in the systemic circulation, including thrombin fragments [48]. At the same time, the thrombin-induced proteolytic activation of PAR1 promotes cell invasion by inducing the persistent transactivation of EGFR and ErbB2/HER2 and, as a result, a prolonged ERK1/2 signaling that initiates tumorigenesis and is involved in the pathogenesis of inflammatory breast cancer [49]. Another perspective is that AF promotes atrial structural remodeling apoptosis through angiotensin II-dependent and angiotensin II-independent pathways [50]; cellular apoptosis, which may result in asymmetric production and disposition in the systemic circulation proapoptotic; and anti-apoptotic molecules, which may lower cancer cell apoptosis and strengthen cancer germinal development [51]. Finally, the medical treatment of AF has been associated in the literature with the occurrence of BC and, more specifically, Amiodarone, which may be associated with an increased risk of incident cancer, with a dose-dependent effect [52].

More specifically, research in this field began in 2008 with an observational study that reported that in newly diagnosed AF cases the frequency of new breast cancer diagnosis was three times higher compared to the general population [53]. Since then, five other studies have been conducted, as well as one control patient study [54] and three cohort studies [55,56,57], which showed that in patients with AF, there is a greater chance of being diagnosed with breast cancer with a noticeable rise in the risk of cancer after 90 days of new-onset atrial fibrillation and a significant decrease 90 days after the first diagnosis. Finally, one study demonstrated an association not with AF but with taking amiodarone for its treatment [58]. It is worth noting that although antiarrhythmic drugs may contribute to atrial fibrillation due to estrogenic effects, the risk of breast cancer in patients with atrial fibrillation remains significantly higher after adjusting for hormone replacement therapy [58].

### 3.3. Myocardial Infarction and Breast Cancer

#### 3.3.1. The Role of Macrophages in the Post-MI Myocardium

Following MI, several pathophysiological processes lead to myocardial healing and cardiac remodeling. They are achieved through an incredibly intricate process that is shared by molecular, cellular, physiological, and immunological events [59]. These generate structural and functional changes in the heart that, in certain cases, have negative effects on the patients. For instance, acute myocardial infarction (MI)-related left ventricular remodeling [60] increases the risk of heart failure. From an immunological perspective, macrophages appear to play a key role in each of these processes. Macrophages offer two distinct functions in tissue regeneration and myocardial injury, respectively [61]. Macrophages make up a sizable group of cells in the case of myocardial infarction, including resident tissue macrophages (rtMacs), resident cardiac macrophages (rcMacs) [62], and monocyte-derived macrophages [63]. Different cell populations are prepared to function in specific ways on the myocardium. rtMacs are self-renewing cells that protect the myocardium after MI by multiplying without relying on feedback from circulating monocytes. The function of monocyte-derived macrophages, which engulf the ischemic tissue and either cause tissue healing or harm, is antidiametrical.

More specifically, it was thought that the only big phagocytes involved in tissue repair were bone marrow-derived macrophages [64]. This idea is debunked by contemporary research, which emphasizes the wide-ranging functions of resident tissue macrophages, particularly about the control of the inflammatory response after injury and in tissue remodeling [62]. The early studies, which primarily used animal models and were conducted in the 1990s on liver tissues [65,66,67,68], focused on separating macrophage populations. This differentiation was later expanded to include the colonization of the heart [69]. It is possible to separate macrophages into at least two different subsets, the CCR2 (+) (C–C chemokine receptor type 2) population and the CCR2 (−) population, based on their cardiac origin and placement. While CCR2 (+) cells, which come from fetal monocyte progenitors and are thought to be of primitive hematopoietic origin and are seen in the trabecular projection of the endocardium, are thought to originate from the yolk sac and are primarily found in the myocardial wall near the coronary vasculature [70]. CCR2 (−) rMacs are additionally referred to as “resident populations” due to their embryonic origin and inherent capacity for self-renewal.

Human, mouse, and rat macrophages can be distinguished using the CCR2 (+/−) marker. Other markers that can be used in combination for humans and rats are the C-X-3-C Motif Chemokine Receptor 1 (CX3CR1) and the major histocompatibility complex class II (MHCII), CD68 [71,72,73], MerTK (myeloid-epithelial-reproductive tyrosine kinase) [74], Mac-3 [75], galactose-specific lectin 3 (Galectin 3) [76], and CD163 [72]. Lymphocyte antigen 6 complex locus C (Ly6C) and MHCII have been used exclusively in murine hearts [77,78], whereas for human samples, the equivalent marker is CD14 [79]. The Ly6C pathway with the CD14 analog in humans has particularly interested researchers both for its action in the myocardium and for the systemic extensions at the organismal level. CD14 (+) CD16 (−) monocytes are classical mediators of inflammation, analogs of non-resident monocytes, and, in mice, constitute the population of Ly-6Chigh monocytes [77]. In contrast, CD14+CD16+/Ly-6Clow cells are associated with rcMacs persisting in the circulation; however, their role in inflammation is less clear [79] and these cells patrol the endothelium and may play a role in immune surveillance [36,80,81].

#### 3.3.2. The Role of Macrophages in the Pathogenesis of Breast Cancer

Macrophages have a catalytic role in the initiation of breast cancer, the growth of the tumor, and the incidence of metastases [82]. There are different immunohistochemistry phenotypes of macrophages that are derived from resident tissue macrophages or are derived from monocytes in the breast, just like there are in the myocardium. Although both types of cells’ functions have been thoroughly investigated, the majority of the evidence indicates that monocyte-derived macrophages have a greater role in the formation of tumors [82,83,84,85,86]. Nevertheless, investigations have shown that resident macrophages also contribute to the metastatic cascade and the growth of the tumor [87,88].

In the landscape of cancer immunology, macrophages play a crucial role, especially in breast cancer, where their presence often signifies a dire prognosis. Unlike other immune cells, the prognostic implications of tumor-associated macrophages (TAMs) are not straightforward. This is due to their remarkable adaptability, shaped by the tumor microenvironment (TME). “M1-like” TAMs are typically anti-tumoral, releasing inflammatory cytokines and reactive molecules [89], while “M2-like” TAMs facilitate tumor growth through immunosuppressive and tissue remodeling factors [90]. Yet, this binary classification is reductive; breast cancer TAMs exhibit a complex, mixed expression of M1 and M2 markers, reflecting a nuanced spectrum of activation. Some TAMs even promote angiogenesis or the establishment of pre-metastatic niches, adding layers to their functional diversity [91]. The relationship between TAMs and the tumor-specific extracellular matrix (ECM) is a longstanding one, characterized by a dynamic crosstalk that influences both TAM function and ECM composition. This review will dissect how the ECM shapes TAM behavior in breast cancer and how TAMs, in turn, modify the ECM to benefit tumor proliferation and metastasis. Deciphering these mechanisms offers a promising therapeutic frontier for tackling resistant or poorly immune-infiltrated tumors, potentially revolutionizing treatment strategies for such cancers [92].

However, a look into breast ontogeny is required before examining the macrophage populations in charge of carcinogenesis in the breast. The mammary gland experiences significant changes during a woman’s lifetime, starting with its development in utero [93]. The breast continuously remodels its duct epithelium, milk-producing cells, acinar cells, and duct lobules to be able to respond to changes [94]. For many years [95,96], it was thought that the main macrophages engaged in the aforementioned activities were produced from bone marrow (BM) and colonized the breast after birth. Nevertheless, while the breast is at rest, Angela C. L. Chua et al. in 2010 discovered two populations of macrophages in the mammary gland with CSF1+ F4/80hi and CSF1+ F4/80 concentrated in the ductal epithelium [94]. Later researchers used fate-mapping in animal models to corroborate the embryonic resident population, with the definitive proof identifying the place of origin of the cells from the yolk sac [97].

With these findings, it is evident that resident macrophages have colonized the breast; yet, during remodeling, tissue growth, or inflammatory conditions, BM-derived monocytes swarm to the gland and partially replace fetal macrophages. According to Jäppinen et al., only the F4/80Int population might be substituted by postnatal [97]. Noteworthy is the fact that resident macrophages in the mammary gland varied in their expression levels of other distinct cell markers, including CD206, CD64, MHC II, MerTK, Siglec-1, Ly6C, Lyve1, and CX3CR1 [98]. This information may one day help us understand the origin of tissue-resident macrophages in the breast tissue in greater detail.

The concept of tumor-associated macrophages (TAMs) is used in place of resident macrophages in the context of breast carcinogenesis. In solid tumors, a population of invading or residing macrophages that exhibit some heterogeneity is referred to as TAMs [99]. TAMs are involved in carcinogenesis and tissue remodeling [100]. There are TAMs made from circulating monocytes in addition to embryonic TAMs; they are referred to as the CD45+CD11b+F4/80+CD206+ population [101]. However, fate-mapping investigations revealed that fetal-derived macrophages in the breast displayed distinct CD206hi compared with postnatal BM-derived monocytes, indicating that some TAMs had embryonic origins [101].

### 3.4. Discussion

It is clear that there is a causal relationship between CVDs and breast cancer. Many studies approach the topic with reverse onco-cardiology as the connection between the two. However, the data we receive from the literature presents an inconsistency. While there is an abundance of observational studies associating breast cancer with previously established HTN and AF, no basic research studies focus on investigating their pathogenetic association. On the other hand, in the case of myocardial infarction, there are animal models that explain the role of activated macrophages and their association with breast cancer, but there are no clinical observational data in humans.

Another serious weakness of the existing literature is that it does not study carcinogenesis in specific organs in isolation, but most studies report populations that developed tumors regardless of location. So, we have mixed data from studies that were not specifically designed for the breast. This work is the first systematic review on BC and CVDs and its purpose is to highlight the need for a multisystem approach to CVD patients about the underlying risk of mammary tumorigenesis. Reverse onco-cardiology in the field of the breast is still in its infancy, and more studies, exclusively designed for the breast, are required to safely answer the questions of correlation between the two pathological entities. If the scientific community answers the challenge, then we will have direct applications in clinical practice that may benefit a large proportion of women. CVDS prevention and breast cancer screening are routine clinical practices, but should not they be approached independently?

From our own study, it appears that the existing literature has not progressed to an analytical approach to the causal relationship between breast cancer and cardiovascular events. Studies approaching the case of myocardial infarction are limited exclusively to data from animal experiments, while in the case of hypertension and AF, the vast majority of studies were not exclusively designed to study the correlation of the two pathologies. From this we understand that on the one hand there are many observations but on the other hand further data are necessary to document a scientific conclusion.

## 4. Conclusions

This review highlights the intricate connection between CVD and breast cancer, emphasizing the need for a multisystem approach to patient care. The concept of reverse onco-cardiology, particularly concerning breast health, is nascent, and more breast-specific studies are required to confirm the correlations observed. Prevention strategies for CVD and breast cancer screening are routine but could benefit from a more integrated approach considering the potential links between these two health concerns. Further research is necessary to substantiate any scientific conclusions about the causal relationship between cardiovascular events and breast cancer, given the current lack of direct analytical approaches in the literature.

## Figures and Tables

**Figure 1 ijms-24-16500-f001:**
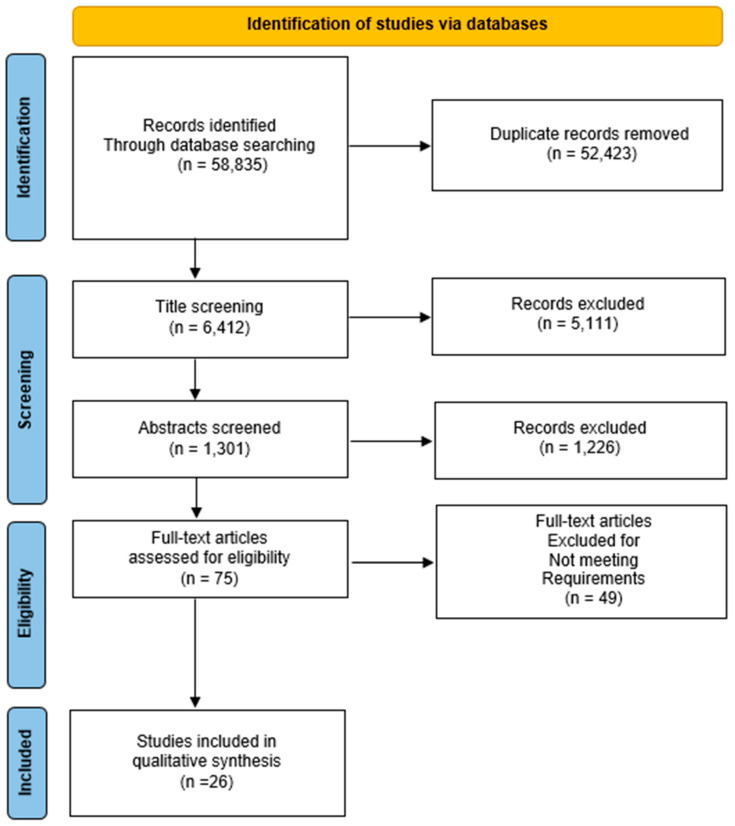
PRISMA diagram showing the screening and processing of articles included in this study.

**Figure 2 ijms-24-16500-f002:**
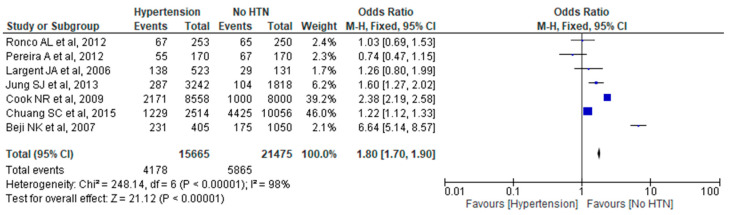
Forest plot indicating statistical significance for hypertension not affecting breast cancer in two groups: hypertension group and control [26,27,28,29,30,31,33].

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
