# Peer review of "Reverse Onco-Cardiology: What Is the Evidence for Breast Cancer? A Systematic Review of the Literature"

_ijms, 2023, doi:10.3390/ijms242216500_

Round 1
Reviewer 1 Report
Comments and Suggestions for Authors
This is a valuable study, but there are some points.
1- Texture editing is necessary.
2- There are many repetitive sentences, especially in the discussion and conclusion, which need to be rewritten
3- The study did not mention and pay attention to the types of breast cancer, especially the hormone-positive type, and the possibility of more connection with CVD.
4- Macrophages are discussed separately in breast cancer and CVD, but no proper analysis has been done in this regard. Also, the inflammatory function and the role of macrophages are not mentioned.
5- The manuscript does not have a suitable conclusion.
Author Response
Dear Reviewer,
Firstly, thank you for your time and effort in reviewing our work. Please find in red, our responses to your comments below.
This is a valuable study, but there are some points.
- Texture editing is necessary.
We performed texture editing as recommended, thank you for the comment.
- There are many repetitive sentences, especially in the discussion and conclusion, which need to be rewritten
We apologise for the conclusions part, we re-wrote the section.
- The study did not mention and pay attention to the types of breast cancer, especially the hormone-positive type, and the possibility of more connection with CVD.
Thank you very much for the comment. There are no studies studying the contrasting effect of cvds on specific histological types of breast cancer. What you mention is about cvd development after hormone dependent cancers which is not the subject of this study
- Macrophages are discussed separately in breast cancer and CVD, but no proper analysis has been done in this regard. Also, the inflammatory function and the role of macrophages are not mentioned.
A sentence was added in macrophages section to elucidate the role of macrophages in breast cancer development according to their inflammatory action ‘’in the landscape..such cancers’’
- The manuscript does not have a suitable conclusion.
The conclusion section has been re-written.
Kind Regards,
The authors
Reviewer 2 Report
Comments and Suggestions for Authors
Dear Authors,
Thank you for your appreciable study. I can not find any positive interplay between betablocker usage and increased breast cancer incidence according to your described cause and effect mechanism, actually accordingly the incidence of co-occurrence should be decrescendo.
The conclusion part of the article has been duplicated exactly.
Author Response
Dear Reviewer,
Firstly, thank you for your time and effort in reviewing our work. Please find in red, our responses to your comments below.
Dear Authors,
Thank you for your appreciable study. I can not find any positive interplay between betablocker usage and increased breast cancer incidence according to your described cause and effect mechanism, actually accordingly the incidence of co-occurrence should be decrescendo.
The conclusion part of the article has been duplicated exactly.
Dear reviewer, thank you for your comment. We added a sentence describing the possible tumor causative relationship between the b blockers and breast cancer ‘The link between..transition’. We also would like to apologise for the conclusions part, we re-wrote the section.
Kind Regards,
The Authors
Reviewer 3 Report
Comments and Suggestions for Authors
Thank You for the beautiful review.
The introduction is concise and catchy.
The methods are clearly and logically described with a comprehensive figure.
The results are well grouped and the section 3.3.2 is very interesting.
The discussion and conclusions are brief and well written.
I have few suggestions:
- I assume the references in the introduction (Lines 39-46) could be placed in the classical manner (numerical).
Line 150 - please expand the hypertension abbreviation (HTN) first used.
Line 176 - I propose name "Beta-blockers", or a greek, not latin letter.
Line 177 - the second beta-blocker is not clear
Line 187 - DBP (I assume, diastolic blood pressure) to be expanded.
Line 215 - atrial fibrillation (AF) abbreviation to be mentioned.
Line 246 - Subsection should be "3.3.1" I assume
Lines 364-366 - maybe this paragraph could be moved to discussion.
Lines 376-378 - good, but repeated phrase as in discussion. Maybe it could be slightly changed or left to one section only.
Author Response
Dear Reviewer,
Firstly, thank you for your time and effort in reviewing our work. Please find in red, our responses to your comments below.
Thank You for the beautiful review.
The introduction is concise and catchy.
The methods are clearly and logically described with a comprehensive figure.
The results are well grouped and the section 3.3.2 is very interesting.
The discussion and conclusions are brief and well written.
I have few suggestions:
- I assume the references in the introduction (Lines 39-46) could be placed in the classical manner (numerical).
Thank you very much for the comment we have changed it accordingly.
Line 150 - please expand the hypertension abbreviation (HTN) first used.
Thank you for noticing, we have amended this now.
Line 176 - I propose name "Beta-blockers", or a greek, not latin letter.
Thank you for the suggestion. We have changed it all and used the greek letter beta ‘β’.
Line 177 - the second beta-blocker is not clear
It was a typo , thank you very much for flagging it.
Line 187 - DBP (I assume, diastolic blood pressure) to be expanded
Thank you for noticing. This has changed accordingly to be more clearer.
Line 215 - atrial fibrillation (AF) abbreviation to be mentioned.
Thank you very much for the comment we have changed it accordingly.
Line 246 - Subsection should be "3.3.1" I assume
Thank you very much for the comment we have changed it accordingly.
Lines 364-366 - maybe this paragraph could be moved to discussion.
Thank you for your suggestion.
Lines 376-378 - good, but repeated phrase as in discussion. Maybe it could be slightly changed or left to one section only.
Thank you for your comment. The Conclusion and Discussion sections were re-written hopefully to your liking. For any additional feedback please do let us know.
Kind Regards,
The Authors
Reviewer 4 Report
Comments and Suggestions for Authors
Dear authors,
I find your manuscript very interesting. Here are my suggestions:
- On lines 41-42 the authors state: "19.05 million CVD-related deaths were 41 anticipated in 2020, an increase of 18.71% from 2010". Since we are in 2023, why talk about what was anticipated rather than what actually happened?
- Also, on lines 42 and 46, the references should be formatted as they currently are written with names rather than numbers.
- The quality of Figures 1 and 2 should be improved.
- The discussions section and the conclusions section are identical! Please check and fix this.
Author Response
Dear Reviewer,
Firstly, thank you for your time and effort in reviewing our work. Please find in red, our responses to your comments below.
Dear authors,
I find your manuscript very interesting. Here are my suggestions:
- On lines 41-42 the authors state: "19.05 million CVD-related deaths were 41 anticipated in 2020, an increase of 18.71% from 2010". Since we are in 2023, why talk about what was anticipated rather than what actually happened?
You are absolutely right and thank you for noticing. We have updated with data from 2021.
- Also, on lines 42 and 46, the references should be formatted as they currently are written with names rather than numbers.
Thank you for the comment. We have now changed them accordingly.
- The quality of Figures 1 and 2 should be improved.
We have redone both figures, hopefully they are clearer now.
- The discussions section and the conclusions section are identical! Please check and fix this.
Thank you for your comment. We have now rewrote both the Conclusion and Discussion sections. If you have any additional feedback please do let us know.
Kind Regards,
The Authors
Round 2
Reviewer 1 Report
Comments and Suggestions for Authors
There are not any extra comments.